# Current Situation and Perspectives of Fruit Annonaceae in Mexico: Biological and Agronomic Importance and Bioactive Properties

**DOI:** 10.3390/plants11010007

**Published:** 2021-12-21

**Authors:** Luis M. Hernández Fuentes, Efigenia Montalvo González, Maria de Lourdes García Magaña, Luis M. Anaya Esparza, Yolanda Nolasco González, Zuamí Villagrán, Sughey González Torres, José Joaquín Velázquez Monreal, David Antonio Morelos Flores

**Affiliations:** 1Instituto Nacional de Investigaciones Forestales, Agrícolas y Pecuarias, Santiago Ixcuintla 63300, Nayarit, Mexico; nolasco.yolanda@inifap.gob.mx; 2Laboratorio Integral de Investigación en Alimentos, Tecnológico Nacional de México/Instituto Tecnológico de Tepic, Tepic 63175, Nayarit, Mexico; mgarciam@ittepic.edu.mx (M.d.L.G.M.); luis.aesparza@academicos.udg.mx (L.M.A.E.); daanmorelos@ittepic.edu.mx (D.A.M.F.); 3División de Ciencias Agropecuarias e Ingenierías, Centro Universitario de los Altos, Universidad de Guadalajara, Tepatitlán de Morelos 47620, Jalisco, Mexico; 4División de Ciencias Biomédicas, Centro Universitario de los Altos, Universidad de Guadalajara, Tepatitlán de Morelos 47620, Jalisco, Mexico; blanca.villagran@academicos.udg.mx (Z.V.); sgonzalez@cualtos.udg.mx (S.G.T.); 5Instituto Nacional de Investigaciones Forestales, Agrícolas y Pecuarias, Tecomán 28925, Colima, Mexico

**Keywords:** Annonaceae, *Annona* species, postharvest technologies, nutritional quality, bioactive compounds, biological activity, perspectives

## Abstract

The Annonaceae family is one of the oldest angiosperms. The genus *Annona* is the one with the most species and, together with *Asimina,* the only ones that contain edible fruits. In the last 10 years, interest in these fruit species has increased, mainly due to their nutritional properties and their application in the treatment of human diseases. Mexico is the center of origin for most of them. However, at present much of the basic agronomic information, postharvest handling of the fruits, and their potential as new crops for areas with poor soils in organic matter or semi-dry climates is unknown. It is considered that these custard apple species may be an option to change towards instead of crops that have lost profitability and sustainability. A review of the current state of knowledge in different areas of the species *A. muricata, A. macroprophyllata, A. reticulata, A. squamosa*, and *A. cherimola* was carried out and to focus research efforts on the topics of greatest interest and on those where is required to achieve a sustainable production and use of these resources in Mexico. However, knowledge about the cultivation and potential uses of these species is needed to increase their commercialization; the integration of interdisciplinary and interinstitutional groups is required.

## 1. Introduction

The Annonaceae family is one of the oldest angiosperms. In the analysis of fossils of leaves, seeds, fruits, and pollen, it is estimated that it had its origin in the late Cretaceous period [1]. The Annonaceae family is significant for ecological, evolutionary, and economic reasons [2]. Worldwide, the Global Biodiversity Information Facility (GBIF) database lists 162 genera, and 3049 species of the Annonaceae family, distributed mainly in the tropical and subtropical region of Central and South America, Africa, Asia, and Australia [3]. On the other hand, the World Checklist of Vascular Plants (WCVP) lists 111 genera with 2444 accepted species [4]. The most representative genera of the family are *Annona* with 215 species, *Goniothalamus* with 147 species, *Guatteria* with 314 species, *Polyalthia* with 153 species, and *Uvaria* with 247 species, and *Xylopia* with 216 species [3], which together represent 42.3% of the species of this family.

In this context, in North America, the Integrated Taxonomic Information System (ITIS) includes 14 genera and 35 species. While the database of the Sistema Nacional de Información de la Biodiversidad (SNIB) mentions the presence in Mexico of 22 genera and 73 species, of these, 19 belong to the genus *Annona* [5]. Only the genera *Annona* and *Asimina* contain species that produce edible fruits; most of these species are found in *Annona* (*A. cherimola* Mill., *A. squamosa* L., *A. muricata* L., *A. cherimola* x *A. squamosa*, *A. reticulata* L., *A. macroprophyllata* Donn. Sm., *A. glabra* L., and *A. purpurea* Moc. & Sessé ex Dunal, while only *A. triloba* is present in *Asimina* [6].

It is estimated that the genus *Annona* originated approximately 66 million years ago in the tropics and subtropics of America, Southeast Asia, and Africa [7,8]. *A. cherimola, A. squamosa, A. macroprophyllata (= A. diversifolia),* and *A. muricata* are native to Central America, and were distributed by humans to other countries [6,9]. *A. muricata*, *A. reticulata*, *A. squamosa*, and *A. cherimola* may have been cultivated by the Mayans for at least 3400 years B.C., to which they made selections during their cultivation [10]. On the other hand, it is believed that *A. reticulata* is native to western India and was later carried to Central America [11]. *A. cherimola* is a species cultivated mainly in the tropics and subtropics of America. Unlike the other fruit species of *Annona*, it requires a temperate climate; it is cultivated in the United States, Mexico, Spain, Australia, Guatemala, Ecuador, and Peru [6,8], Spain being the main producer with 3600 hectares. In Mexico, its development and agronomic knowledge are still limited; 36 hectares have been established, with a production of 247.3 tons [12]. Of the five fruit species of *Annona*, *A. muricata* currently presents greater agronomic development and established surface. It is cultivated mainly in Mexico, Venezuela, Ecuador, Colombia, and Brazil in areas with a warm climate from sea level to an altitude of 800 m. It is native to Central America [1,9]. Currently, Mexico and Brazil are the main producers, and together they maintain around 8000 cultivated hectares [12,13]. On the other hand, *A. reticulata* and *A. macroprophyllata* are still species with little or no cultivation development. Both are currently in a semi-cultivated state in backyard orchards or in the wild in tropical and warm regions in the Mexican Pacific states. *A. macroprophyllata* is native to Mexico. This species was a fruit known and consumed by the Aztecs before the arrival of the Spanish, who brought it to Europe in the year 1570 [14]. It is mainly distributed in the dry tropic region in the states of Guerrero and Colima [15]. It is commercialized in local markets; the fruits are collected in wild areas, from backyard trees, or established on the banks of other crops, such as corn, and mainly grazing lands. Efforts have been made to identify and characterize phenotypes of this species with outstanding adaptive and agronomic characteristics, highlighting the collections and characterizations of Ballesteros in the Tierra Caliente region, in the state of Guerrero [16,17,18]. It tolerates soil conditions with low levels of nutrients, drought, and high temperatures, which makes it an interesting alternative species to grow in places with these conditions and contributes to the economy of low-income families [16]. *A. reticulata* is found in the wild in Mexico; however, in other countries such as India, Brazil, Peru, Bangladesh, Pakistan, and the United States, it is found in small areas of cultivation, in backyards, and in the wild from where it is collected for its consumption and sale in local markets [11]. *A. squamosa* is a little better-known species; it is cultivated in Central America, Mexico, India, and northwest Brazil [12,19,20]. It is distributed in tropical and subtropical countries in warm and semi-warm places from 200 m to 1300 m above sea level [21]. Brazil is the main producer, with around 10,000 hectares cultivated in the northwest [13].

The use of fruit Annonaceae is wide and varied; these fruits are consumed principally fresh. The processed fruits can be found in the market as different anonas-derived products such as ice cream, flakes, juice, nectar, jelly, yogurt, puree, milkshakes, syrup, jam, alcoholic beverages, desserts, and extracts of active compounds with medicinal properties [22]. However, during the last two decades, these *Annona* species have been the subject of numerous and diverse studies mainly due to their biological properties (ecology) and applications in the fields of medicine (various pathologies), nutraceuticals (antioxidants, minerals, and fiber), metabolomics (acetogenins and other insecticidal, fungicidal, and bactericidal compounds), and agriculture (alternative crops, molecular characterization, germplasm collection, characterization, plant breeding, and obtaining new varieties), among others [23].

Due to their genetic characteristics and biotechnological potential, interdisciplinary research groups have been formed in countries such as Spain, France, the United States, Japan, Brazil, Argentina, Colombia, and Mexico [24], and recently India and Bangladesh, as highlighted by numerous publications. In Mexico, the National Network of Anonaceas (NNA) was formed in 2002, made up of researchers from different institutions and disciplines of knowledge, with an emphasis on the study of the species *A. muricata*, *A. cherimola*, *A. squamosa*, *A. diversifolia*, *A. reticulate*, *A. purpurea*, and related wild genera that, faced with the increasing threat of environmental deterioration, are at risk of genetic erosion and gradual loss. The main objectives are the generation of knowledge for in situ and ex situ conservation, use and enhancement, and capacity building in these species [25]. Although some activities have been carried out in the NNA, the lack of budget support has limited the monitoring and fulfillment of these objectives. Currently, this network is part of the subcommittee on genetic resources, which in turn is coordinated by the Sectoral Committee on Genetic Resources for Food and Agriculture, whose legal basis is on the Agreement by which the Sectoral Committee on Genetic Resources for Food and Agriculture is created. The objective of this Committee is to promote the conservation, management, fair and equitable distribution of benefits, and sustainable use of genetic resources, through inter-institutional and interdisciplinary coordination in the sector [26]. Similarly, specialist meetings have been held with presentations of research results in different areas of knowledge in the Annonaceae group [24,25].

Considering the above, the objective of this review is to gather the current information generated in these *Annona* species, delimit the frontier of knowledge, and propose actions in the short, medium, and long term for the NNA. The JSTOR, ResearchGate, Google Scholar, Scopus databases and national production statistics, as well as technical publications from research institutes and centers were reviewed.

It is interesting to observe how the development of research in these Annona species has been, on the one hand, the basic agronomic knowledge for their commercial production, and postharvest management technologies and obtaining derived products is still limited. They have made progress only in some areas of the production process and postharvest handling. However, in topics such as phytochemistry, nutraceutical uses, and applications in different pharmacological areas, important research contributions have been constructed in several countries.

## 2. Agronomic Knowledge

Of the species of fruit custard apples or soursop, *A. muricata* is the one with the greatest development in its agronomic knowledge in Mexico. It has the largest cultivated area and production [12]. In Mexico, clonal varieties of *A. muricata* (breeder titles: Guanay-1, Guanay-2, and Guanay-3) have been characterized and registered [27]. However, most of the current agronomic knowledge is generalized. More specific studies are required, such as adaptation in different environments, generation of varieties with higher yields and tolerance to pests and diseases, studies of phenology and nutritional requirements, planting density and formation pruning, collection, morphological and genetic characterization, and genotype conservation. Currently, there is greater knowledge in different areas of biology, agronomy, and biotechnology in this species. Its agronomic management can be consulted in [18,28,29,30,31,32]. Some specific work has been carried out which contributes to optimal production. The management of the main pests and disease problems can be consulted in [28,33,34,35,36]. Similarly, important advances have been made in the collection, description, and selection of genotypes [32,37,38,39], with which improvement activities could begin.

The species *A. cherimola* is second in economic importance in Mexico. However, the cultivation area and production are smaller, with an establishment potential in central and southern Mexico [40]. Currently, it is only cultivated in Michoacán and Morelos [12]. It is found naturally in temperate and subtropical zones in the states of Morelos, Oaxaca, Michoacán, Hidalgo, Veracruz, Chiapas, Puebla, Jalisco, Guanajuato, and the State of Mexico [41]. In this context, some genotypes of *A. cherimola* have been characterized and registered, including Tonaltzintl, Metztli 34, Selección 94-33, Álvaro, Urhuata, and Lamtl 256 [27]. As with *A. muricata*, its agronomic knowledge in Mexico is limited, and the recommended management is generalized [42,43]. Research has been carried out in specific areas such as pollination [44], selection and characterization [45,46], and pests and pathogens [47,48,49,50]. The knowledge of the crop is more advanced in Spain, technology has been developed, along with a selection of varieties for their best use [51,52,53,54,55].

On the other hand, the species *A. macroprophyllata, A. reticulata,* and *A. squamosa* have less development and knowledge in Mexico. These species do not have a breeder’s title and registration in National System of Genetic Resources (SNICS- acronym in Spanish), and there is less agronomic knowledge about them. However, they have a potential for cultivation as an alternative or complementary option to the production of tropical fruit trees that have lost profitability. *A. macroprophyllata*, commonly known as ilama, is found naturally from Colima in Mexico to El Salvador [14]. At present, there is no knowledge of any cultivated area, as its trade is local, and all of the fruit comes from the collection of trees that are found naturally on the edge of pastures or orchards and in backyards. The first studies for its knowledge in Mexico were carried out by members of the National Network Anonaceae (NNA). In the communities of Tierra Caliente, in the Balsas region of southern Mexico, this fruit is most widely known. Its production is rustic; it presents tolerance to high temperatures, drought, and soils poor in nutrients and organic matter [16]. A research group from the Technological Institute of Ciudad Altamirano, Guerrero, as members of the NNA, has managed to identify and describe the physical-chemical characteristics of 24 types of ilama, of which 146 accessions were collected for propagating and promoting its conservation and cultivation. It is estimated that there are approximately 100 types of ilama in this region [16]. The morphological description and genomic identification of the most representative types of ilama can be consulted in the catalog Diversity in the ilamas of Tierra Caliente del Balsas, Mexico [56]. In addition to the above, more specific studies have been carried out to identify its chemical composition and antioxidant activity [57]. This species represents an interesting option due to its adaptation and pleasant flavor, with development potential for the semi-dry climate regions in the states of Colima, Guerrero, Michoacán, and Oaxaca.

*A. reticulata* is known as sugar apple, custard apple, or ox heart custard apple; native to India and Central America, there are no reports about the large of cultivated areas. It is mainly found naturally in tropical and subtropical regions, and only a few small areas are cultivated in south Florida, Bahamas, Bangladesh, Pakistan, and some parts of India [58,59]. In Mexico, there are no reports of its cultivation. It is found in its natural state in the Pacific region, from southern Sinaloa to Chiapas; it is not cultivated commercially, and its consumption is local in the production season. It has been observed in Nayarit, Jalisco, and Colima that the fruits are collected in the months of March to May; their consumption is fresh, and presents a great phenotypic variety, which would imply greater selection, characterization, and possible use as a crop in these areas. Its potential for use and research in medicine has great potential; numerous studies have been carried out on its chemical constitution and pharmacology, finding satisfactory results [60,61,62,63,64]. Regarding *A. squamosa*, known as saramuyo, it has a cultivated area of 50.5 hectares in Mexico, with a production of 431 tons. Its cultivation is recent, and it is located mainly in the south of the country, in the state of Yucatán. [12]. It is distributed naturally from the tropics in Mexico to Bolivia [21,65], with Brazil being the main producer [13]. Until a few years ago, in Mexico, the saramuyo was only found to be naturally occurring; however, recent research has contributed to a better understanding of this species. Similar to other Annona species, in saramuyo, there are numerous studies on chemical composition, evaluation, and use of its pharmacological properties [66,67,68], and little research on the selection, characterization, improvement, and agronomic management. General recommendations for its knowledge and agronomic management can be consulted in the Manual of Propagation and production of saramuyo (*Annona squamosa* L.) [69]. However, its potential for alternative cultivation as the main activity in areas with warm and semi-dry climates and soils poor in the organic matter makes the saramuyo a species that requires more attention and research.

### 2.1. Nutritional Composition

Table 1 shows the nutritional composition and energy value of the edible part of *A. muricata, A. cherimola, A. squamosa,* and *A. macroprophyllata*. *A. cherimola* and *A. squamosa* fruits are sweeter than *A. muricata*, *A. reticulata,* and *A. macroprophyllata*. They have a low content of lipids and proteins [70,71,72,73,74,75,76,77,78,79]. Furthermore, the pulp exhibits a good quantity of dietary fiber, although *A. muricata* and *A. cherimola* fruits are more fibrous. The pulps are prominent in some vitamins (C, E, thiamin, riboflavin, and niacin) and minerals (Ca, P, Mg, K, and Fe). In addition, they are very low in calories. In this context, the regular consumption of these fruits provides essential nutrients for the recommended daily intake.

### 2.2. Tradicional Medicine Use

In addition to the nutritional importance of the *Annona* fruits, different reports have demonstrated the use of all parts from the trees of *A. muricata, A. cherimola, A. squamosa, A. reticulata,* and *A. macroprophyllata* in traditional medicine. Since many years ago, the native people of various cultures have consumed beverages produced from Annonaceae stem bark, roots, leaves, fruit pulp, peel, and seed for a wide range of illnesses. The preparation of beverages has been by maceration, decoction, or infusion [80]. People have consumed these products for the treatment of illnesses such as parasitic infections, dysentery, fever, urethritis, hematuria, and asthma, liver diseases, diarrhea, and for their anticold, antispasmodic, antisudorific, anti-depressive, and antiemetic properties [81]. Nonetheless, most biological properties of members of the Annonaceae family are attributed to the presence of diverse secondary metabolites/bioactive compounds [82].

### 2.3. Bioactive Compounds Identified in Fruit Annona Species

Table 2 and Table 3 show some phytochemicals (quantitative or qualitative) identified in pulp, leaf, peel, seeds, roots, and stem bark of fruit *Annona* species: *A. muricata, A. cherimola, A. squamosa, A. reticulata,* and *A. macroprophyllata* [83,84,85,86,87,88,89]. These compounds include phenolic compounds, alkaloids, fatty acids, cyclopeptides, alkaloids, and acetogenins.

#### 2.3.1. Phenolic Compounds and Fatty Acids

The phenolic compounds are the most abundant metabolites in *Annona* fruits, and they are considered the principal exogenous antioxidants. Their basic structure contains an aromatic ring with one or more hydroxyl substituents [83]. However, there are some *Annona* species that have not been completely characterized, and one of them is *A. macroprophyllata* (Table 2). In *A. muricata* has been identified and quantified the highest diversity of phenolic compounds from pulp, leaf, peel, and seed compared to those observed in other *Annona* species. Among the most significant are hydroxybenzoic acids, hydroxycinnamic acids, and flavonoids. In a similar way, phenolic compounds have been reported in the pulp and leaves of *A. cherimola*. The presence of flavonoids, for example, derivates of quercetin, gallocatechin, and kaempferol, give significant biological value to fruits as well as leaf extracts [84]. Moreover, the presence of phenolic compounds and flavonoids has been qualitatively reported in the edible part, leaf, and seed of *A. squamosa* [83,84,85,90,91,92,93,94,95,96,97]. Chlorogenic acid, ferulic acid, *p*-hydroxybenzoic acid, caffeic acid, gallic acid, and epicatechin are the most abundant phenolic compounds in this specie [86].

The fatty acids are compounds based on carbon chains with a methyl group at one end of the structure and a carboxyl group at the other end. They are predominantly hydrophobic compounds [79,86]. The seeds from *Annona* fruit are mainly valuable for their fatty acid content (Table 2). For example, the palmitic, stearic, oleic, and linoleic acids were identified as the major fatty acids of *Annona* seed oils. In general, the content and type of phenolic compounds and fatty acids depend on each plant part and fruit *Annona* species; however, their biological importance is due to their contribution to the total antioxidant activity with pharmacological activities, mainly in preventing or slowing the progress of diverse oxidative stress-related diseases when they are consumed [87].

#### 2.3.2. Cyclopeptides

Cyclopeptides or cyclic peptides are polypeptides formed from amino acids organized in a cyclic ring structure [101,102]. In the seeds of some *Annona* species bioactive compounds named cyclopeptides have been reported that in recent years have been of much pharmaceutical interest. Annomuricin A-C has been extracted and identified from *A. muricata* seeds (Table 3) [89,101,102]. Two other cyclopeptides (cherimolacyclopeptide E and F) were obtained from *A. cherimola* seeds [103,104]. Moreover, seven cyclopeptides (cyclosquamosin A-G and met-cherimolacyclopeptide B) were reported in seeds, and two cyclopeptides (fanlizhicyclopeptide A and B) were found in the peel of *A. squamosa* [105,106,107,108]. The *Annona* cyclopeptides show a remarkable variety of derivatives and a wide array of bioactivities, including antiproliferative, antimicrobial, anthelmintic, and anti-inflammatory, and cytotoxicity [104].

#### 2.3.3. Alkaloids

Alkaloids that have been qualitatively identified in different plant parts of the *Annona* species are the isoquinoleic alkaloids. The most frequently found include aporphine, protoberberines, benzyloquinoleins, and the bis-benzylisoquinolein dimers [83]. In the leaves, seeds, stem bark, root, and peel from *A. muricata*, *A. cherimola*, *A. squamosa,* and *A. diversifolia*, anomurine, anomuricine, atherosperminine, coreximine, coclaurine, corytenchine, isocoreximine asimilobine, annonaine, corypalmine, liriodenine, nornuciferine, reticuline have been reported, among many others [109,115,119]. The alkaloids with (R)-apomorphine groups have been clinically used for the treatment of Parkinson’s disease [144].

#### 2.3.4. Acetogenins (ACGs)

ACGs are molecules characterized by a long aliphatic chain of 35 to 37 carbon atoms with one, two, or three tetrahydrofuran or tetrahydropyran rings in their central region [124]. Annonaceae acetogenins are secondary metabolites with the highest therapeutic interest, due to their cytotoxic capacity on cancer cell lines and their in vivo antitumoral activity [82]. The acetogenins are found in the seeds (*A. muricata*, *A. cherimola,, A. diversifolia*, *A. squamosa,* and *A. reticulata*), leaves (*A. muricata*, *A. cherimola*, and *A. squamosa*), edible pulp (*A. muricata* and *A. squamosa*), roots (*A. muricata*), and stem bark (*A.*
*muricata, A. cherimola,* and *A.*
*squamosa*). Although there are more than five hundred identified acetogenins, annonacin and bullatacin are considered the acetogenins with the highest concentration in *Annona* species. Moreover, they have excellent cytotoxicity at low doses, but they can be highly toxic if consumed in high concentrations [82,145]. They can be found in the pulp, leaves, seeds, and roots of *A. muricata* [83,124,125,127,134]. In the same way, annonacin and bullatacin have been identified in the pulp and stem bark of *A. squamosa* [135,138]. In *A. macroprophyllata,* the principal reported acetogenins from seeds are rolliniastatin-2, laherradurin, and cherimolin-2 [88,89].

### 2.4. Biological Activities from Extracts or Isolated Compounds from Annona Species

There are many biological activities attributed to extracts (aqueous, ethanolic, methanolic, and ethyl acetate) or isolated compounds from *Annona species* such as anti-angiogenic, analgesic, thrombolytic, anti-ulcer, anti-platelet, vasorelaxant, anti-pyretic, anti-convulsant, anti-depressive, neuroprotective, anxiolytic, antitumoral or cytotoxic, hypoglycemic, and anti-inflammatory [146]. Table 4 shows several studies on the biological effects of extracts or isolated compounds from principal fruit *Annonas.*

#### 2.4.1. Cytotoxic Activity

Cytotoxic activity has been studied in fruit, leaves, root, and stem bark extracts or isolated acetogenins of some *Annona* species (Table 4). Annonacin at a very low dose (extracted from *A. muricata* leaves) decreased the induced-tumors on mice [147,148]; moreover, acetogenins have a potent cytotoxicity against many cancer cell lines (DU-145, A-549, MCF-7, HT-29, A-498, PC-3, HL60, HCT, Caco-2, and HepG-2) [88,96,119,137,143,149,150,151,152]. The mechanism of action of acetogenins on cancer cell lines or in vivo cancer models is attributed to inhibition mitochondrial complex I (ubiquinone-linked NADH oxidase) to present in cell membranes; the ATP decrease cause apoptosis [123].

#### 2.4.2. Anti-Hyperglycemic Activity

Various studies on anti-hyperglycemic effects are reported for *Annona* species. The anti-hyperglycemic activity of *A. squamosa* leaves, roots, and peel extracts have been evaluated in hyperglycemic or diabetic models [153,154,155,156,157,158,159,160,161,162,163]. Table 4 shows that leaves have been studied more than other tissues. The glucose reduction in blood by *Annona* extracts is related to the presence of polyphenols [156].

#### 2.4.3. Anti-Inflammatory

Anti-inflammatory activity has been explored in fruit, leaves, root, and stem bark extracts of some *Annona* species [164,165,166,167,168,169,170], as shown in Table 4. *A. cherimola* leaf extract exhibited a potent in vivo anti-inflammatory activity on induced-inflammation rats [167]. The anti-inflammatory activity of *Annona* extracts is related to the content of some phenolic compounds, alkaloids, kaurenes, or sesquiterpenes in the extracts [169,170].

### 2.5. Technologies for Postharvest Handling

The fruit belonging to the genus *Annona* are climacteric, with respiration rates of up to 350 mL of CO_2_ kg^−1^ h^−1^, in addition to having ethylene production values of 46.2 and 68.5 µL kg^−1^ h^−1^ at a storage temperature of 25 to 30 °C [171]. Due to the above, the production capacity of *Annona* is wasted, since the production of these fruit in the country presents various management problems [172]. Moreover, a 60% post-harvest loss has been reported, which causes the producer to choose to move their fruit in transport with high cost (air) and use expensive post-harvest technologies to delay the ripening of the fruit, mainly refrigeration (13 °C) [173].

During the handling of the fruit, it is essential to preserve the physical, chemical, and organoleptic characteristics that are ideal for the consumer [174], which is why the essential result is to carry out an adequate postharvest handling. However, there are no postharvest technologies registered or issued by organizations for the handling of fruit belonging to the genus *Annona* [175]. The considerations for the post-harvest handling of soursop (*Annona muricata* L.) that are applied are those described by Morton, where it is mentioned that the fruit needs to be kept under conditions of room temperature and relative humidity of 85 and 90%. In addition, it is necessary to be cautious when handling the fruit, since they are susceptible to physical, chemical, and microbiological damage [176]. The common operations in the post-harvest handling of soursop are harvesting, selection, cleaning of organic residues, classification, antifungal treatment, weighing, pre-cooling (12 to 15 °C), drying of residual humidity, post-harvest treatment (waxing), storage, and transport [177].

As mentioned above, high rates of respiration and ethylene production play an important role in the postharvest handling of the genus *Annona*. Ethylene is responsible for regulating the maturation and senescence of various agricultural products [178]; therefore, it is responsible for the fruit to acquire optimal organoleptic characteristics for consumption, as well as for the senescence of the fruit [179]. Eventually, having identified the mechanisms of action of ethylene on the fruit, it has been possible to develop technologies and procedures to control the different negative effects that this hormone can cause (Table 1). In research carried out by Wijesinghe et al., the effect of 1-methylcyclopropene (1-MCP) at 0.3, 0.6, and 0.9 µL/L was evaluated in four different *Annona* fruit: *A. atemoya, A. reticulata, A. muricata,* and *A. squamosa*. As response variables, the days of storage, physiological weight loss (PWL), firmness, color, total soluble solids (TSS), and titratable acidity (TA) were measured. At the end of the investigation, it was concluded that the application of 1-MCP increased the shelf life to three extra days for all the fruit compared to the control fruit without treatment. The 0.6 µL/L concentration reduced PWL, the enzyme activity responsible for the loss of firmness and control of chlorophyll degradation in addition to regular TSS and TA [180].

Other technologies used to extend the shelf life of perishable fruit are coatings or so-called “waxes”. Montalvo-González et al. [181] used candelilla and beeswax emulsions as postharvest technology applied to soursop fruit (*A. muricata*). Respiration rate, ethylene production, PWL, firmness, TSS, TA, pH, and color (°hue) were considered as response variables. The authors describe a favorable effect on respiration rate, since it decreased in the fruit treated with wax and managed to delay the appearance of the climacteric peak one day after the control fruit. A similar effect was found in the rate of ethylene production since the levels of the treated fruit were lower compared to the control fruit. Finally, for TSS, TA, pH, and color, a moderating effect was noted in the generation of these parameters since the metabolic processes of the fruit were not inhibited or modified [181].

By showing an interesting effect on the genus *Annona*, 1-MCP and waxes were studied in combination due to their high performance and potential applications.

The research of Tovar et al. [173] was based on applying to soursop fruit 1000 nL/L of 1-MCP in combination with three different wax formulations: first, carnauba wax type III Emulwax 3060; second, carnauba wax type III, and food-grade silicone oils Emulwax 3061; and third, refined candelilla wax Emulwax 3070 (all formulations were applied after 1-MCP). The determinations that were used as response variables were PWL, TSS, TA, pH, color (° hue), and firmness. In general, the combination of 1-MCP and the three types of coatings managed to lengthen the shelf life of the fruit by an extra six days, as compared to the control (18 days in total). In the PWL analyses, it was discovered that the combination of these technologies managed to significantly reduce this parameter; however, no difference was found between the treatments. Regarding the content of TSS, TA and pH were observed a major delay in the normal evolution of the variables. The values of Hue for the control fruit and combination 1 and 2 did not show significant differences; however, for combination 3, a decrease in color development was noted. The author attributed this phenomenon to the synergistic effect of formulations 1 and 3. Finally, in the firmness data, was reported significant differences between the treated and control fruit, but no differences were found between treatments [173]. Table 5 shows some technologies to extend the shelf life of *Annona* fruits.

## 3. Perspectives

Since the agronomic knowledge of these fruit species of *Annona* in Mexico is still limited, it is necessary to exploit the conditions of the center of origin of most of these and the optimal climatic conditions for their cultivation. Therefore, the Red Nacional de Annonaceae should initially focus its activities on the generation of knowledge of best agronomic practices that allow for obtaining the highest yields and the least environmental impact of these crops to achieve their sustainable development. There is a lack of selected varieties and characterization of the great gene pool that is still found in natural conditions, backyard, or underutilized on the verges of harvesting roads or grazing areas. It is necessary to carry out an exhaustive search, characterize and begin an improvement program with the most outstanding productive genotypes or with some characteristics of interest. The potential for commercial use of these species is underutilized, given the great acceptance of the fruits for fresh consumption, and obtaining products and applications in areas such as medicine or therapeutic treatments. Currently, it is still important to look for extraction alternatives to increase the yield and demonstrate the distribution and contents of compounds in the different *Annona* species. Although these species are rich in secondary metabolites, it is not fully clear if they can be a source of nutraceutical metabolites for the pharmaceutical industry. According to the review, various extracts of *A. muricata, A. cherimola, A. squamosa, A. macroprophyllata,* and *A. reticulata* species contain metabolites capable of exerting different biological activities that are beneficial to health; however, further studies on the precise mechanisms of action and expanding the knowledge of *Annona* species less investigated such as *A. reticulata* are required.

## Figures and Tables

**Table 1 plants-11-00007-t001:** Nutritional composition of edible parts from *Annona muricata, A. cherimola, A. squamosa,* and *A. macroprophyllata* fruits.

Parameter	*A. muricata*	*A. cherimola*	*A. squamosa*	*A. macroprophyllata*
Total Energy (kcal)	55.4–81.73	81–102	92.9–97.7	56.06–89.03
Moisture (%)	80.48–83.2	68.7–70.4	65–75	71.5–79.61
Protein (%)	0.69–1.10	1.36–1.96	0.7–1.89	0.44–1.31
Lipids (%)	0.20–0.97	0.10–0.29	0.048–0.57	0.16–0.31
Soluble carbohydrates (%)	12.50–18.23	13.0–29.0	20.41–25.19	13.55–20.25
Dietary fiber (%)	4.83–5.76	2.09–5.32	0.62–1.41	0.97–1.30
Minerals (mg/100 g)				
Calcium	9.0–10.3	9.0–27.14	17.0–44.7	0.86–31.60
Phosphorous	27.7–29.0	24.0–35.20	54.0	51.7
Iron	0.64–0.82	0.25–0.60	0.3–1.34	ND
Magnesium	22.0	17	21.0–22.0	8.0–14.01
Copper	ND	ND	0.086	ND
Manganese	ND	ND	0.10	ND
Zinc	ND	0.16	0.1	0.10–0.13
Potassium	320	288	142	335.95–347.40
Vitamins (mg/100 g)				
Vitamin C	22.59–40.56	12–6–25.43	25.6–58.75	1.51–13.6
Vitamin E	29	0.27	ND	ND
Thiamin	0.11–2.10	0.09–0–0.11	0.10	0.24
Riboflavin	0.05–0.2	0.11–0.13	0.06	0.30
Niacin	0.21–1.52	0.65–1.0	0.89	2.18
Cobalamin	ND	0.12	ND	ND
**Refences**	[70,71]	[72,73,74]	[75,76,77,78]	[76,79]

ND = No determined.

**Table 2 plants-11-00007-t002:** Bioactive compounds: phenolic compounds and fatty acids from different plant components of some *Annona* species.

*Annona* Specie	Plant Part	Bioactive Compounds	Content	References
		**Phenolic compounds**		
*A. muricata*	Pulp (μg/g dw)	Chlorogenic acid	12.80	[83]
		Cinnamic acid	42.04	
		Coumaric acid	0.07	
		Gallic acid	15.86	
		4-Hydroxybenzoic acid	131.63	
		Neochlorogenic acid	72.32	
		Dicaffeoylquinic acid, caffeoylquinic acid, fisetin, dihydrokaempferol-hexoside, morin, kaempferol 3-*O*-rutinoside, kaempferol, luteolin 3′7-di-*O*-glucoside, myricetin	NR	[90,91,92]
	Leaf (μg/g dw)	Caffeoylquinic acid, chlorogenic acid, dicaffeoylquinic acid, feruloylquinic acid, cinnamic acid, isoferulic acid, caffeic acid, gallic acid, apigenin-6-C-glucoside, argentinine, catechin, coumaric acid, daidzein, epicatechin, gallocatechin, genistein, glycitein, homoorientin, kaempferol, kaempferol 3-*O*-rutinoside, luteolin 3′7-di-*O*-glucoside, quercetin, quercetin 3-*O*-glucoside, quercetin 3-*O*- neohesperidoside, quercetin 3-*O*-robinoside, quercetin–O-rutinoside, quercetin 3-*O*-α-rhamnosyl, robinetin, tangeretin, taxifolin (+),vitexin	NR	[90,91,92,98]
	Peel (μg/g dw)	Gallic acid	14.50	[83]
		Coumaric acid	1.37	
		Cinnamic acid	45.51	
		Caffeic acid	43.68	
		Chlorogenic acid	32.67	
		Protocatechuic acid	150.46	
		4-Hydroxybenzoic acid	145.98	
		Syringic acid	883.71	
		Neochlorogenic acid	78.86	
	Seed (μg/g dw)	Gallic acid	0.36	[83]
		Coumaric acid	0.07	
		Cinnamic acid	40.48	
		Caffeic acid	32.62	
		Chlorogenic acid	12.33	
		Protocatechuic acid	133.47	
		Syringic acid	780.77	
		Neochlorogenic acid	69.70	
*A. cherimola*	Pulp	Catechin, procyanidin (B-type) dimer isomer, (epi)catechin-(epi)gallocatechin, epicatechin, derivates of procyanidin trimer	NR	[93]
	Leaf (mg/100 g dw)	Catequin	12.42–24.5	[84]
		Quercetin 3-*O*-rutinoside-7-*O*-glucoside	1.06–16.16	
		Epicatechin	6.33–26.31	
		Quercetin 3-*O*-rutinoside-7-*O*-pentoside	32.25–75.29	
		Quercetin 3-*O*-rutinoside	719.53–2593.92	
		Kaempferol-3-Galactoside-7-Rhamnoside	55.74–620.98	
		Kaempferol-3-*O*-glucoside	22.44–337.09	
		Luteolin-3-Galactoside-7-Rhamnoside	47.49–120.58	
*A. squamosa*	Leaf	Quercetin, quercetin 3-*O*-glucoside, rutin, gallic acid, chlorogenic acid, isorhamnetin, ferulic acid, kaempferol, caffeic acid	NR	[85,95]
	Seed	Gallic acid, ρ-hydroxybenzoic acid, syringic acid, ferulic acid, ellagic acid, benzoic acid, o-coumaric acid, and salicylic acid	NR	[96]
*A. diversifolia*	Peel (mg/100 g)	Chlorogenic acid	84	[86]
		Ferulic acid	34.9	
		p-hydroxybenzoic	20.90	
		Caffeic acid	6.6	
		Gallic acid	4.9	
		Epicatechin	102	
	Seed (%)	Oleic acid Palmitic acid Linoleic acid Stearic acid	70.42 16.40 7.97 5.22	[79]
		Palmitic acid		
*A. muricata*	Seed (%)	Linoleic acid	25.5	[97]
		Stearic acid	1.5	
		Stearic acid	6.0	
		Oleic acid	39.5	
		Linoleic acid	27.0	
*A. cherimola*	Seed (%)	Myristic acid	0.05	[99,100]
		Palmitic acid	14.91	
		Heptadecanoic acid	0.21	
		Stearic acid	7.60	
		Palmitoleic acid	0.32	
		Oleic acid	35.20–43.72	
		Linoleic	32.48–44.93	
*A. squamosa*	Seed (%)	Oleic acid	41.9	[87]
		Palmitic acid	14.7	
		Linoleic acid	26.6	
		Stearic acid	11.3	
		Hexadenoic acid	10–14	
		Hepatadecene-(8)-carbonic acid (1)	29.68	
		Cis-vaccenic acid	10.39	
		Heneicosanoic acid	3.20	
		9-octadecenoic acid (Z)-,2,3-dihydroxy propyl ester	13.33	

NR = Not reported.

**Table 3 plants-11-00007-t003:** Bioactive compounds: cyclopeptides, alkaloids, and acetogenins from different plant components of some *Annona* species.

*Annona* Species	Plant Part	Bioactive Compound	References
		**Cyclopeptides**	
*A. muricata*	Seed	Annomuricatin A-C	[101,102]
*A. cherimola*	Seed	Cherimolacyclopeptide E, herimolacyclopeptide F	[103,104]
*A. squamosa*	Seed	Cyclosquamosins A–G,	[105,106,107]
		cyclosquamosin, met-cherimolacyclopeptide B	
	Peel	Fanlizhicyclopeptide A-B	[108]
		**Alkaloids**	
*A. muricata*	Root, bark	Anomurine, anomuricine, atherosperminine, coreximine, coclaurine	[109]
	Stem	Atherospermine, casuarine, 2,5-dihydroxymethyl-3,4,dihydroxypyrrolidine, deoxymannojirimycin, deoxynojirmycin	[109,110]
	Leaf	Stepharine, coclaurine, coreximine, annonaine, asimilobine, 2,5-dihydroxymethyl-3,4,dihydroxypyrrolidine, deoxymannojirimycin, deoxynojirmycin, swainsonine, (R)-O,O-dimethylcoclaurine, annonamine, R)-4′O-methylcocaurine, S)-narcorydine, xylopine, N-methylcoclaurine, remerine, isoboldine, isolaureline, liriodenine, reticuline, N-methylcoculaurine	[109,110,111,112,113,114]
	Fruit	Nornuciferine, annonaine, asimilobine	[111,113,114]
	Peel	Nornuciferin, assimilobin, anonaine, isolaureline	[83]
*A. cherimola*	Leaf	Liriodenine, anonaine, nornuciferine, 1,2- dimethoxy-5,6,6a,7-tetrahydro-4 h, dibenzoquinoline-3,8,9,10-tetraol, asimilobine, pronuciferine	[84]
	Root	Corytenchine, isocoreximine	[115]
*A. squamosa*	Leaf	Reticuline, o-methylarmepavine, annonaine, oxophoebine, lysicamine, n-methylcoclaurine, liriodenine, corydine, lanuginosine, roemerine, corypalmine, sanjoinine, norlaureline, norcodeine, oxalanobie, aporphine	[75,116,117,118]
	Seed	Annonaine, asimilobine, liriodenine, corypalmine, reticuline, nornuciferine	[119]
	Stem bark	Roemerolidine, *N*-nitrosoxylopine, duguevalline	[120]
*A. diversifolia*	Seed	Rolliniastatin-2, laherradurin, cherimolin-2 and liriodenine	[88,121]
	Root, Stem bark, leaf	Liriodenine, atherospermidine, lysicamine	[122]
		**Acetogenins**	
*A. muricata*	Pulp	Montecristin, epomuricenins A-B, epoxymurin, epomurinins A-B, epomusenins A-B, annonacin, corossolone, muricatin C, muricin, montanacin	[123,124,125,126]
	Leaf	Annocatalin, annohexocin, annomuricin A-E, annomutacin, annonacin, annopentocin A-C, corossolone, gigantetronenin, goniothalamicina, montanacin, muricapentocin, muricatalicin, muricin, muricatalin, muricatocin A-C, murihexocin, muricoreacin, solamin	[82,127,128,129]
	Seed	Ronbusticin, annomuricatin A, cohibin A-D, donhexocin, muricatenol, murihexol, epomuricenins A-B, corepoxylone, Epoxyrollin B, annoglaxin, annomontacin, annonacin, annoreticuin-9-one, arianacin, corossolina, corossolona, gigantetrocin A-B, goniothalamicina, muricatetrocin A-B, Muricatin A-D, muricin A-I, murisolina, solamin, bullatacilin, bullatacin, gigantecin, annocatacin	[82,89,101,130,131,132,133,134]
	Root	Cohibin A-B, montecristin, epomuricenins A-B, sabadelin, annonacin	[82,125]
	Stem bark	Muricatin C	[124]
*A. cherimola*	Pulp	Bullatacin, annonacin	[135]
	Leaf	Molvizarin, cherimolin-1, motrilin, annonacin, annonisin	[136]
	Seed	Cherimolin, dihydrocherimolin, molvizarin, motrilin, itrabin, jetein, cherimolin-2, almunequin, annomolin, annocherimolin	[137,138]
	Stem bark	Aromin-A, squamocin	[138]
	Leaf	Murihexocin C	[139]
*A. squamosa*	Seed	Annotemoyin, squamocin, annoglaxin, epoxyrolin, murisolin, neo- desacetyluvaricin, squamostatin, annosquamins, bullatacin, annosquacin, annosquatin, annonareticin, motrilin, solamin, squadiolin, squamoxinone, squamostanin, uvarigrandin, squamostolide, tripoxyrollin, uvariamicin	[75,140,141,142]
	Stem bark	4-deoxyannoreticuin, *cis*-4-deoxyannoreticuin, (2-4-*cis* and *trans*)-squamoxinone, annoreticuin-9-one, bullatacin, molvizarin, mosin, parviflorin, squamotacin	[75,143]
*A. macroprophyllata*	Seed	Laherradurin, rolliniastin-2, and cherimolin-2	[89]

**Table 4 plants-11-00007-t004:** Cytotoxicity, anti-hyperglycemic, and anti-inflammatory activities of different crude extracts or isolated compounds from different plant parts of some *Annona* species using different model assays.

Activity/*Annona* Species	Plant Part	Bioactive	Dose/Concentration	Model Assay	Effect	References
**Cytotoxicity**						
*A. muricata*	Leaf	Annonacin	85 nM	Mice induced a skin tumorogenesis	To reduce the tumor incidence, tumor burden, and tumor volume	[147]
		Acetogenin-rich fraction	100 and 200 mg/kg for 7 days	Rats induced benign prostatic hyperplasia (PSA)	The fractions (200 mg/kg) significantly reduced the PSA level	[148]
	Root-bark, fruit, leaf	Ethyl acetate extract	50 µg/mL	MCF-10A cell line (breast)	The fractions had the highest anticancer abilities	[149]
	Stem bark	Annonacin and ethyl acetate extract	IC_50_ 0.1 μM and 55.501 μg/mL	DU-145 prostate carcinoma cells	Annonacin and extract displayed selective and potent cytotoxicity	[150]
*A. cherimola*	Seed	Annomolin, annocherimolin		A-549 (lung), MCF-7 (breast), HT-29 (colon), A-498 (kidney), PC-3 (prostate) and MIA PaCa-2 (pancreas) cell lines	Potent cytotoxicity against all cell lines	[137]
		Ethanolic extract	IC_50_ 23.20 μg/mL	Colorectal cancer cell lines: T84, HCT-15, SW480 and HT-29, cancer stem cells	Potential cytotoxic activity on T-81 and HCT-15 resistant cell lines	[151]
	Leaf	Ethanolic extract	IC_50_ 390.20 μg/mL	Breast cancer cell lines: MDA-MB-231 and MCF-7	Selective antiproliferative and pro-apoptotic activities	[152]
*A. squamosa*	Stem bark	Bullatacin	IC_50_ of 2.47 × 10^−7^ µg/mL	A-549, HT-29, MCF-7, A-498, PC3 and PACA-2 cell lines	Selective cytotoxic activity against MCF-7 cells	[142]
	Seed	Acetogenins	IC_50_ ranged from 2.2 × 10^−1^ to 8.3 × 10^−3^ µg/mL	HeLa, MCF-7, A-549, Hep-G2, SMMC-7721 and MKN-45 cell lines	Selective cytotoxic activity against MCF-7 and A-549 cells	[141]
	Leaf	Methanolic extract	IC_50_ ranged from 1.1 to 2.1 µg/mL	Human immortalized line of T lymphocyte (Jurkat), MCF-7, HL60, and HCT-116	Leaf extract was more active against MCF-7 cells, likewise for seed extract against Jurkat and HL60 cells	[119]
	Seed, peel, pulp	Aqueous extract	IC_50_ 7.31 ± 0.03 and 15.99 ± 1.25 µg/mL	Cancer cell lines colon (Caco-2), prostate (PC3), liver (HepG-2), and breast (MCF-7)	Seed extracts had the lowest IC_50_ values for PC-3 and MCF-7 cancer cell lines	[96]
*A. macroprophyllata*	Seed	Cherimolin-2	IC_50_ of 0.5 µg/mL for SW-480 and for HeLa 0.05 µg/mL. In vivo doses 500 mg/kg body weight, 20 days	HeLa and SW-480 cell line and rats injected with HeLa (1 × 10^6^) or SW-480 (5 × 10^6^) cells	Size reduction of HeLa tumor (43%), and 16% of SW-480 tumor	[88]
		Laherradurin	IC_50_ of 0.15 µg/mL for HeLa and SW-480. In vivo doses 500 mg/kg body weight 20 days	HeLa and SW-480 cell line and rats injected with HeLa (1 × 10^6^) or SW-480 (5 × 10^6^) cells	Size reduction of HeLa tumor (64%) and SW-480 tumor (60%)	[88]
**Hypoglycemic activity**						
	Leaf	Aqueous extract	100 mg/kg for 2 weeks	Streptozotocin-induced diabetic rats,	Significant reduction of blood glucose levels and protective action on pancreatic β-cells	[153]
*A. muricata*		Dry extract	100 mg/kg/day for 4 weeks	Streptozotocin-induced diabetic rats.	The dry extract improves behavioral alterations and protects testis in diabetic animals	[154]
	Steam bark	Ethanolic extract	150 and 300 mg/kg for 2 weeks	Alloxan-induced diabetic male albino rats		[155]
*A. cherimola*	Leaf	Ethanolic extract	300 mg/kg, one week	Alloxan-induced type 2 diabetic (AITD)	Attenuated postprandial hyperglycemia	[156]
		Aqueous extract (infusion) (1.5 g/mL)	300 mg/kg	Streptozocin-induced diabetic mice	Reduction of the blood glucose level, glycated hemoglobin, cholesterol, and triacylglycerols	[157]
*A. squamosa*	Leaf	Aqueous extract	300 mg/kg body weight	Streptozotocin-induced Wistar rats	Decreased glucose and increased insulin sensitivity	[158]
		Ethanolic extract	100 mg/kg body weight	Streptozotocin-induced rats	Glycemia decreased as well as glycated hemoglobin, creatinine, and urea	[159]
	Root	Aqueous extract	500 mg/kg body weight	Streptozotocin-induced hyperglycemic rats	A significant reduction of glycemia, 6 h after oral administration	[160]
	Leaf and peel	Aqueous extract	250 mg/kg body weight	Streptozotocin-induced diabetic rats	Improvement of the glycemia and lipid profile	[161]
*A. macroprophyllata*	Leaf	Ethanolic extract	200 mg/kg body weight	Alloxan-induced diabetic Balb-c mice	A significant decrease in postprandial hyperglycemia	[162]
		Aqueous extract	300 mg/kg body weight	Healthy rats	Reduction of glycemia at a dose-dependent manner	[163]
**Anti-inflammatory activity**						
*A. muricata*	Fruit	Aqueous extract	50, 100 and 200 mg/kg	Carrageenan-induced paw edema rats and xylene-induced-ear edema rats	Significant anti-inflammatory activity on paw edema and ear edema	[164]
	Leaf	Aqueous extract	100 mg/kg body weight	Female Balb/c albino mice injected with *Escherichia fergusonii*	The extract minimized the inflammation by decreasing the expression levels of IL-1β and TNF-α	[165]
		Ethanolic extract	100, 200, and 400 mg/kg orally for 7 days	Rectoanal tissue from Swiss mice	All three doses show significant anti-inflammatory effects on hemorrhoidal tissue	[166]
*A. cherimola*	Leaf	Ethanolic extracts	100 mg/kg	Rats (Leukocyte migration to the peritoneal cavity and Subcutaneous air pouch test)	The inhibitory effect of the ethanolic extract on leukocytes migration was 63.8 and 73.16%	[167]
*A. squamosa*	Root	Ethanolic extract	400 mg/kg body weight	Carrageenan-induced paw edema rats	Inhibition (54%) of edema inflammation at 400 mg/kg	[168]
	Stem bark	Caryophyllene oxide	25 mg/kg body weight	Acetic acid-induced Swiss albino rats	Inhibition of inflammation (75% ) at 25 mg/kg	[169]
		18-acetoxyent-kaur-16-ene	25 mg/kg body weight	Acetic acid-induced rats	Inhibition of inflammation 62%) at 25 mg/kg	[170]

**Table 5 plants-11-00007-t005:** Postharvest technologies or treatments to extend the shelf life of fruit of the genus *Annona*.

Technology/Treatment	Genus *Annona*	Effect	References
Aqueous extract of coconut mesocarp + commercial chitosan	*Annona muricata* L.	Control of *Rhizopus stolonifera*	[182]
Electrolyzed solution with neutral pH	*Annona muricata* L.	Sporicidal activity	[183]
Hydrothermal, fungicidal, and wax treatment	*Annona muricata* L.	Browning control	[184]
Edible mucilage coating	*Annona muricata* L.	Preservation of the physicochemical characteristics and extending the shelf life (2 extra days)	[185]
1-Methylcyclopropene	*Annona muricata* L.	Preservation of physicochemical characteristics	[186]
Chitosan coating	*Annona muricata* L.	Anthracnose control	[187]
Chitosan coating	*Annona muricata* L.	Control of physicochemical, microbiological, and sensory characteristics	[188]
Cold storage	*Annona muricata* L.	Increase the days of shelf life of the fruit (2 extra days)	[189]
*Bacillus atrophaeus* strain B5	*Annona muricata* L.	Biocontrol of postharvest anthracnose	[190]

## Data Availability

The information presented here is the authors’ revision. The tables and figures are self-made based on the information and review carried out.

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
