# Peer review of "Current Situation and Perspectives of Fruit Annonaceae in Mexico: Biological and Agronomic Importance and Bioactive Properties"

_plants, 2021, doi:10.3390/plants11010007_

Round 1
Reviewer 1 Report
It is opinion of the reviewer that this interesting and well prepared paper before acceptance Leeds several corrections. My individual comments are listed below.
20 – What does it mean “chemical properties”? Correction is needed.
Page 3 A map with the geographical distribution of Annonaceae should be completed.
219 – It should be “edible parts” instead of “edible portion”.
Table 1 – It should be :Lipids” instead of “Fat”.
Table 1 – In footnote, explanation of “ND” is needed.
232 – It should be “secondary metabolites/bioactive compounds”.
249 - – It should be “phenolic acids” instead of “simple phenolic compounds”.should be reported in two separated tables.
263 - The chemical structure of cyclopeptides should be described.
Phenolic compounds and fatty acids
Table 2 – The unit (μg/g dw) should be located in the “Content” column.
Table 2 – For leaf – Firs should be listed all the phenolic acids, then flavonoids.
Table 2 – For the names of flavonoids, “-O-“ must be written with italic.
Table 3 – The chemical names must be written with lower case letters.
248 – The chemical structure of acetogenins should be described.
301 – How the extracts were obtaine?
Table 4 – 85 nM means “85 nmol/L” this is concentration not a dose.
Table 4 – for A. cherimola steam bark, it should be “Ethanolic extract”.
15 – It should be “Ethanolic extract” not “Ethanolic extracts”.
325 – “to protect …characteristics” – wrong sentence.
328 – It should be “technologies” instead of “methodologies”/
345 – It should be “1-methylcyclopropane”.
Table 4 – A term of “treatment should be added to technology#1, 5, 9.
Reviewer 2 Report
The paper presents current situation and perspectives of fruit Annonaceae in Mexico, their biological and agronomic importance and properties. In the last years, interest in these fruit has increased mainly due to their chemical properties and application in the treatment of human diseases. However, at present much of the basic agronomic information, postharvest handling of the fruits and their potential as new crops for areas with poor soils in organic matter is unknown. The review is interesting, however there are some points that need to be clarified or modified before publication.
1) The Authors provided too few keywords.
2) Introduction is too long and should be shortened.
3) Organization of the manuscript is wrong. It should be prepared according to “Instructions for Authors”. There are a lot of typos and errors in the article. Latin names of plant species should be written in italics (e.g. line 206: Annona squamosa L.). Information about Author contributions, sources of funfing, conflicts of interest e.t.c. are necessery.
4) Many items in the literature are out of date. Please, supplement the Refrerences with new scientific reports.
5) Authors should familiarize themselves with the proper format for References and make appropriate corrections.
Reviewer 3 Report
The manuscript from Fuentes et al reviews the literature on the Annona genus an under-utilised fruit-bearing plant. The review collects information on the horticulture of the plants and the phytochemicals they produce. The results of some biological assays are collected as well. The manuscript appears to comprehensively summarize the literature and should be a useful resource for those working with this genus. My only criticism is the English needs some impreovement. Some suggested corrections are listed below.
line 22 replace "theme" with "them"
line 29 Needs to be reworded. "the importance of these species knowledge is scarce" does not make sense
line 34 replace "angiosperms, using" with angiosperms. Using"
line 43 replace ". Which" with ", which"
line 51 replace "Dunal," with "Dunal),"
line 56 delete "for"
line 59 replace "America, unlike" with "America. Unlike"
line 61 replace "being Spain" with "Spain being"
line 65 replace "surface, it" with "surface. It"
line 70 replace development, both" with "development. Both"
line 93 replace "beverages desserts" with "beverages, desserts"
line 102 replace "highlighting" with "and"
line 101 replace "Bangladesh by" with "Bangladesh, highlighted by"
line 108 replace "loss, the" with "loss. The"
line 115 replace "created, the" with "created. The"
line 127 replace "production, postharvest" with "production, and postharvest"
line 127 replace "and" with "while"
line 134 replace "a larger" with "the largest"
line 137 replace "generalized, more" with "generalized. More"
line 144 replace "diseases" with "disease"
line 153 Needs to be reworded. Its not clear what has been characterized and who has registered genotypes.
line 168 replace "shores" with "edge" or "verge"
line 171 replace "known, its" with "known. Its"
line 175 replace "in order to" with "for"
line 176 replace "cultivation, it" with "cultivation. It"
line 185 replace "areas, it" with "areas. It"
line 188 replace "cultivation, it" with "cultivation. It"
line 190 replace "season, it" with "season. It"
line 196 italicize "A. squamosa"
line 196 replace "an area" with "a cultivated area"
line 197 replace "tons, its" with "tons. Its"
line 207 replace "warm climates, semi-dry" with "warm, semi-dry climates"
line 210 replace "nutrimental" with "nutritional"
line 214 replace "Besides, the" with "The"
Table 1 replace "Cooper" with "Copper"
Table 1 the Vitamin E entry for A. squamosa is missing
line 224 replace "in the traditional" with "in traditional"
line 225 replace "have been consumed" with "have consumed"
line 227 replace "beverages have" with "of beverages has"
line 227 replace "decoctions or infusions" with "decoction or infusion"
line 228 replace "considering many health benefits such as: parasitic," with "for the treatment of illnesses such as parasitic infections,"
line 235 replace "screening of" with "identified in"
lines 243 and 244 need to be revised. These lines state that "A. muricata pulp ... has the highest diversity of phenolic compounds", but in Table 2 there are more compounds listed from the peel, leaf and seed. Nearly all of the compounds found in the peel have greater concentrations than in the pulp.
line 251 replace "as" with "are"
line 253 replace "Respect to fatty acids the seeds from fruit Annona" with " The seeds from Annona fruit"
line 253 replace "by" with "for"
line 257 replace "contributed" with "contribute"
line 258 replace "also" with "and"
line 264 replace "much interesting since of the point of view pharmaceutical" with "been of much pharmaceutical interest"
line 264 replace "Have been extracted and identified Annomuricin A-C" with "Annomuricin A-C has been extracted and identified"
line 265 replace "Others two" with "Two other"
lines 267,268 replace "have a special attention by their remarkable variety of cyclopeptide derivates" with "show a remarkable variety of derivatives"
line 283 replace "interesting" with "interest"
line 287 replace "to exist" with "there are"
line 289 replace "to" with "at"
line 290 replace "find" with "be found"
line 293 replace "While" with "In"
line 301 replace "principal fruit Annonas" with "Annona's plant parts"
line 307 replace "with" with "to"
line 314 replace "ad" with "and"
line 345 which hormone is being discussed here"?
line 363 replace "paraments" with "parameters"
line 366 replace "Tovar et al. It was" with "Tovar et al was"
line 371 What is meant by "stay"?
line 378 replace "noted, the author" with "noted. The author"
line 379 replace "1- and formulation 3" with "formulations 1 and 3"
line 382 replace "improvement" with "extend the shelf"
line 385 replace "The" with "Because the"
line 386 replace "condition" with "conditions"
line 392 replace "shores" with "verges"
line 397 replace "looking " with "to look"
line 399 replace ". ." with "."
line 401 replace "for" with "for the"
line 404 replace "expand" with "expanding"
line 405 replace "few" with "less"
lines 406-426 The example text in these lines needs to be replaced with the appropriate information.
